# VirSorter: mining viral signal from microbial genomic data

Simon Roux[1,*], Francois Enault[2,3], Bonnie L. Hurwitz[4] and Matthew B. Sullivan[1,*]

[1] Ecology and Evolutionary Biology, University of Arizona, USA
[2] Clermont Université, Université Blaise Pascal, Laboratoire "Microorganismes: Génome et Environnement," Clermont-Ferrand, France
[3] CNRS UMR 6023, LMGE, Aubière, France
[4] Department of Agricultural and Biosystems Engineering, University of Arizona, USA
[*] Current affiliation: Department of Microbiology, The Ohio State University, USA

Corresponding author
Matthew B. Sullivan,
mbsulli@email.arizona.edu

## ABSTRACT

Viruses of microbes impact all ecosystems where microbes drive key energy and substrate transformations including the oceans, humans and industrial fermenters. However, despite this recognized importance, our understanding of viral diversity and impacts remains limited by too few model systems and reference genomes. One way to fill these gaps in our knowledge of viral diversity is through the detection of viral signal in microbial genomic data. While multiple approaches have been developed and applied for the detection of prophages (viral genomes integrated in a microbial genome), new types of microbial genomic data are emerging that are more fragmented and larger scale, such as Single-cell Amplified Genomes (SAGs) of uncultivated organisms or genomic fragments assembled from metagenomic sequencing. Here, we present VirSorter, a tool designed to detect viral signal in these different types of microbial sequence data in both a reference-dependent and reference-independent manner, leveraging probabilistic models and extensive virome data to maximize detection of novel viruses. Performance testing shows that VirSorter's prophage prediction capability compares to that of available prophage predictors for complete genomes, but is superior in predicting viral sequences outside of a host genome (i.e., from extrachromosomal prophages, lytic infections, or partially assembled prophages). Furthermore, VirSorter outperforms existing tools for fragmented genomic and metagenomic datasets, and can identify viral signal in assembled sequence (contigs) as short as 3kb, while providing near-perfect identification (>95% Recall and 100% Precision) on contigs of at least 10kb. Because VirSorter scales to large datasets, it can also be used in "reverse" to more confidently identify viral sequence in viral metagenomes by sorting away cellular DNA whether derived from gene transfer agents, generalized transduction or contamination. Finally, VirSorter is made available through the iPlant Cyberinfrastructure that provides a web-based user interface interconnected with the required computing resources. VirSorter thus complements existing prophage prediction softwares to better leverage fragmented, SAG and metagenomic datasets in a way that will scale to modern sequencing. Given these features, VirSorter should enable the discovery of new viruses in microbial datasets, and further our understanding of uncultivated viral communities across diverse ecosystems.

## INTRODUCTION

Viruses of microbes, mainly infecting bacteria and archaea, are ubiquitous and abundant in every type of biome sampled thus far, where virus-host interactions alter ecosystem function ranging from geochemical cycling to human health (*Fuhrman, 1999*; *Wommack & Colwell, 2000*; *Weinbauer, 2004*; *Breitbart & Rohwer, 2005*; *Edwards & Rohwer, 2005*; *Suttle, 2007*; *Rohwer & Thurber, 2009*; *Letarov & Kulikov, 2009*; *Rodriguez-Valera et al., 2009*; *Reyes et al., 2012*; *Brum & Sullivan, 2015*). In the oceans, for example, viruses infecting cyanobacteria kill approximately 3% of their hosts per day (*Suttle, 2002*), while also impacting cyanobacterial photosynthesis locally and globally through the expression and transfer of virus-encoded photosystem core genes (*Lindell et al., 2005*; *Sullivan et al., 2006*). Such modulation of host microbial metabolisms during infection appears to be a generalized strategy wherein oceanic viral communities encode genes with the potential to modulate key microbial carbon, nitrogen, phosphate and sulfur metabolisms (*Breitbart et al., 2007*; *Sharon et al., 2009*; *Sharon et al., 2011*; *Thompson et al., 2011*; *Hurwitz, Hallam & Sullivan, 2013*; *Anantharaman et al., 2014*; *Roux et al., 2014b*; *Hurwitz, Brum & Sullivan, 2015*). In humans, viruses of microbes appear dynamic (*Reyes et al., 2010*; *Pride et al., 2011*; *Minot et al., 2013*), and again likely play key ecosystem roles, particularly affecting virulence of facultative pathogens (*Boyd, 2012*; *Busby, Kristensen & Koonin, 2013*) with a striking example being the requirement of a phage infection for the full virulence of *Vibrio cholerae* (*Waldor & Mekalanos, 1996*). Microbial viruses may also help fight antibiotic-resistant pathogens, leading to a recent resurgence in research exploring the use of viruses for "phage therapy" in humans (*Bush et al., 2011*; *Nobrega et al., 2015*).

In spite of this importance, our understanding of viral diversity remains limited to a tiny fraction of that occurring in nature. This is because most microbes known to exist from barcode surveys are not yet in culture (*Rappé & Giovannoni, 2003*), and even if microbial hosts were cultivated, not all viruses are amenable to cultivation (*Edwards & Rohwer, 2005*). In the oceans alone, the lack of reference genomes leads to surveys of viral communities returning mostly (63–93%) unknown sequences (*Brum & Sullivan, 2015*), and most (99%) of 5,476 surface ocean viral populations remaining taxonomically unidentifiable beyond the "order" level (*Brum et al., 2015*). This is not surprising, given that 86% of the 1,531 genomes of viruses that infect bacteria and archaea available at RefSeq are associated with only 3 of 61 known host phyla (based on the viral genomes available in NCBI RefseqVirus v69, January 2015).

One way forward is to better detect and catalog viral sequence data from rapidly expanding microbial genomic datasets. First, prophages, which result from the integration of a temperate virus genome into a microbial host genome, are present in ∼60% of sequenced bacteria (*Casjens, 2003*; *Canchaya, Fournous & Brüssow, 2004*). Second, Single-cell Amplified Genome (SAG) datasets are now routinely generated to provide

genome sequence data and inferences about metabolic capacity for novel microbes (*Swan et al., 2011*; *Kamke, Sczyrba & Ivanova, 2013*; *Rinke et al., 2013*; *Kashtan et al., 2014*), and offer a rich source of novel viral sequences. These data will include prophage sequences, as well as viruses from actively lytic infections. Such SAG-based viral signal has already provided insights into marine viral diversity and virus-host interactions in uncultivated protists, bacteria and archea (*Yoon et al., 2011*; *Roux et al., 2014b*; *Labonté et al., 2015*). Third, large genome fragments of uncultivated microbes and associated viruses can now be assembled from microbial metagenomes (*Sharon et al., 2009*; *Sharon et al., 2011*; *Narasingarao et al., 2012*; *Albertsen et al., 2013*; *Anantharaman et al., 2014*). Finally, viral metagenomics (viromics) can be used to survey the sequence data associated with purified viral particles and can also result in assembly of large viral genome fragments (*Emerson et al., 2012*; *Minot et al., 2013*; *Roux et al., 2013*; *Brum et al., 2015*).

Numerous approaches are available to identify prophages in complete microbial genomes including Phage_Finder (*Fouts, 2006*), Prophinder (*Lima-Mendez et al., 2008*), PHAST (*Zhou et al., 2011*), and PhiSpy (*Akhter, Aziz & Edwards, 2012*). Overall, prophage predictors rely on the detection of sequence similarities between regions of the microbial genome and known viral genes. In addition, PhiSpy also identifies "viral-like" genomic features (AT and GC skew, protein length and transcription strand directionality) to enable the detection of viruses absent from databases (*Akhter, Aziz & Edwards, 2012*). Prophage predictors also look for prophage "ends" by identifying the attachment sites in the microbial genome for each predicted prophage. These tools are either designed for a user to download and run locally (PhiSpy, Phage_Finder) or to access through a web-server (PHAST).

However, new tools are needed that (i) advance viral detection beyond prophages and instances where new viruses closely match those available in databases, and (ii) can handle fragmented and larger-scale microbial genomic datasets. Here, we present VirSorter, an automated tool designed to detect viral signal in genomic datasets, and make this new tool and the associated databases freely available in the Discovery Environment of the iPlant Cyberinfrastructure (*Goff et al., 2011*). Overall, we demonstrate that VirSorter detects prophages in complete microbial genomes as well as current prophage tools, but also offers capabilities to detect viral sequences in fragmented genomic datasets including incomplete genomes, SAGs or metagenomic assemblies, and can be used to flag potential cellular contamination in viromes for removal.

## MATERIALS & METHODS

### Building reference databases for bacterial and archaeal viruses

Two reference databases of viral protein sequences were built for VirSorter and are available in the iPlant Discovery Environment (Data/Community_Data/iVirus/VirSorter/Database). The first includes 114,297 proteins from viruses infecting bacteria or archaea in RefSeqVirus genomes (as of January 2014), hereafter named "RefSeqABVir." Protein clusters (PCs) were defined using MCL clustering (*Enright, Van Dongen & Ouzounis, 2002*) of these proteins (inflation 2.0) based on their reciprocal blastp comparisons (threshold of

50 on bit score and $10^{-03}$ on E-value). The 9,735 PCs with at least 3 sequences were used to define a profile database searchable with HMMER3 tools (*Eddy, 2011*). The remaining 34,668 unclustered sequences were formatted for a blastp search. All PCs that did not contain any sequences from *Caudovirales* and unclustered sequences from viruses other than *Caudovirales* were marked as "Non-*Caudovirales*."

The RefSeqABVir database was then augmented by virome sequences sampled from freshwater, seawater, and human gut, lung and saliva, resulting in an extended version of the reference database (hereafter named "Viromes") which includes both virome and RefSeqABVir sequences. This combined reference dataset should help to detect new viruses for which no cultivated reference sequence is available. When only raw reads were available, viromes were assembled using Newbler (threshold of 98% identity on 35bp). The resulting contigs were then checked for the presence of cellular genome sequences, and only the 68 viromes for which no 16S rRNA genes were retained (see Table S1 for a complete list of these viromes). Contigs assembled from these 68 viromes were then manually inspected (through annotations generated by Metavir; *Roux et al., 2014a*) and revealed no identifiable cellular genome sequences (i.e., no sequence contained more than 2 genes that matched a cellular genome and were not found in any known virus). A total of 146,521 complete predicted proteins from this quality-controlled dataset were then clustered with the 114,297 proteins from RefSeqABVir, leading to 15,673 clusters with 3 sequences or more, and 88,052 unclustered sequences. PCs from the combined Viromes database were used to create a profile database searchable with HMMER3, and the 34,338 unclustered sequences from RefseqABVir were formatted for BLAST search (unclustered sequences from viromes were not added to the database to prevent the inclusion of contaminating sequences).

Within these databases, viral "hallmark" genes were defined though a text-searching script looking for "major capsid protein," "portal," "terminase large subunit," "spike," "tail," "virion formation" or "coat" annotations. After a manual curation step removing genes with more general annotation such as "protease" or "chaperone," 826 PCs or single genes were identified as "viral hallmark genes." This latter point meant removing domains also matching "protease" or "chaperone" domains and was conducted to minimize false positives for our viral hallmark genes category by extra-cautiously avoiding PCs that might include domains that could derive from either both viruses or microbes.

## VirSorter sequence pre-processing

VirSorter was inspired by previous algorithms and tools developed to detect prophages (viral sequences integrated in cellular genomes), especially Prophinder (*Lima-Mendez et al., 2008*). For each (set of) genome(s) and/or contig(s) (for draft genomes) provided as raw nucleotide sequences, the initial stages of VirSorter include a detection of circular sequences (i.e., sequences with matching ends likely representing circular templates; *Roux et al., 2014a*), gene prediction on each sequence with MetageneAnnotator (*Noguchi, Taniguchi & Itoh, 2008*), and selection of all sequences with more than 2 genes predicted. VirSorter also removes all poor-quality predicted protein sequences (predicted protein sequences with more than 50 consecutive X, F, A, K or P residues) likely originating from

gene prediction across low-complexity or poorly defined genome regions (e.g., "bridges" between contigs generated during scaffolding) and yielding false-positive matches when compared to protein domain databases.

Predicted protein sequences are then compared to PFAM (v27) and to the viral database selected by the user (either RefSeqABVir or Viromes) with hmmsearch (*Eddy, 2011*) and blastp (*Altschul et al., 1997*) and each gene is affiliated to its most significant hit based on alignment score. Thresholds for significant hits are as follows: minimum score of 40 and maximum E-value of $10^{-05}$ for hmmsearch, and minimum score of 50 and maximum E-value of $10^{-03}$ for blastp.

## VirSorter metrics computation

Following the sequence pre-processing, viral regions are detected through computation of multiple metrics using sliding windows. The metrics used are (i) presence of viral hallmark genes (*Koonin, Senkevich & Dolja, 2006*; *Roux et al., 2014b*), (ii) enrichment in viral-like genes (i.e., genes with best hit against the viral reference database, either RefSeqABVir or Viromes), (iii) depletion in PFAM affiliated genes, (iv) enrichment in uncharacterized genes (i.e., predicted genes with no hits either in PFAM or the viral reference database), (v) enrichment in short genes (genes with a size within the 10% shorter genes of the genome), and (vi) depletion in strand switching (i.e., change of coding strand between two consecutive genes).

For all the enrichment and depletion metrics, a score comparable to the one of Prophinder was used (*Lima-Mendez et al., 2008*). First, a global value for each metric is estimated for the whole genome set (global rate of viral-like genes, global rate of PFAM-affiliated genes, *etc*). Then, for each window, the number of observed events (e.g. number of viral-like genes) is compared to an expected number deduced from the global value of the metric (modeled with a binomial law). A *p*-value is computed, reflecting the probability of observing *n* events or more (for enrichment) or *n* events or fewer (for depletion) at random, thus corresponding to a risk of generating false positives. These *p*-values are multiplied by the total number of comparisons (here the total number of sliding windows observed on a sequence), and a negative logarithmic transformation ($-\log_{10}$) defines the associated significance score, again as in the Prophinder algorithm (*Lima-Mendez et al., 2008*).

For the detection of viral-like genes enrichment, two different values are computed for each dataset: one based on genes in the entire database (RefSeqABVir or Viromes), and another based on non-*Caudovirales* genes only. Indeed, *Caudovirales* genomes represent 81% of RefSeqABVir, and the remaining viral families usually have only a handful of reference genomes. The global rate of viral-like genes in cellular genomes is thus usually one order of magnitude lower when considering only non-*Caudovirales* genes (viral-like genes ratio across the bacterial and archaeal class for which complete genomes are available at NCBI RefSeq and WGS ranges from 4.8 to 16%, with an average of 10.6%, whereas the ratio of non-*Caudovirales* genes in these same genomes ranges from 0.01 to 1.4%, with an average of 0.16%). Hence, the same number of genes in a region would be

considered as non-significant when matching *Caudovirales* (compared to the global rate of *Caudovirales*-like genes in the whole genomes), but would be significant when only composed of non-*Caudovirales* genes.

## Sequence metrics summary

Each metric is computed using sliding windows from 10 to 100 genes wide, starting at every gene along the sequence, and all scores greater than 2 are stored. Local maxima of significance score are then searched and the associated set of genes is defined as a putative viral region. These different predictions (based on the metrics above) are then merged when overlapping (extending the regions to include all predicted windows), leading to a list of putative viral regions associated with a (set of) metric(s). These regions are classified into three categories: (i) *category 1* ("most confident" predictions) regions have significant enrichment in viral-like genes or non-*Caudovirales* genes on the whole region and at least one hallmark viral gene detected; (ii) *category 2* ("likely" predictions) regions have either enrichment in viral-like or non-*Caudovirales* genes, or a viral hallmark gene detected, associated with at least one other metric (depletion in PFAM affiliation, enrichment in uncharacterized genes, enrichment in short genes, depletions in strand switch); and (iii) *category 3* ("possible" predictions) regions have neither a viral hallmark gene nor enrichment in viral-like or non-*Caudovirales* genes, but display at least two of the other metrics with at least one significance score greater than 4. Finally, if a predicted region spans more than 80% of predicted genes on a contig, the entire contig is considered viral. A summary of VirSorter detection types is displayed in Fig. 1B.

Next, higher confidence predictions are used to refine the sequence space search. Specifically, sequences from all open reading frames from *category 1* predictions that do not match a viral protein cluster are clustered and added to the reference database (RefSeqABVir or Viromes depending on the initial user choice). This updated database is then used in another round of search by VirSorter. This iteration where *category 1* sequences are used to refine the searches is continued until no new genes are added to the database. Once no new genes are added, the final VirSorter output is provided to the user and includes nucleotide sequences of all predicted viral sequences in fasta files, an automatic annotation of each prediction in genbank file format, and a summary table displaying for each prediction the associated category and significance scores of all metrics. By providing the predictions and the underlying significance scoring, users can evaluate each prediction and apply custom thresholds on significance scores through a simple text-parsing script, even for large-scale datasets.

VirSorter is available as an application (App) in the iPlant discovery environment (https://de.iplantcollaborative.org/de/) under Apps/Experimental/iVirus (see Fig. S1 for a step-by-step guide of VirSorter app on iPlant). This application allows users to search any set of contigs for viral sequences using either the RefSeqABVir or the Viromes database. The reference values of VirSorter metrics will be evaluated on the complete set of input sequences, hence mixed datasets should be sorted (when possible) by type of bacteria or archaea in order to get the most accurate result possible. In addition to these reference

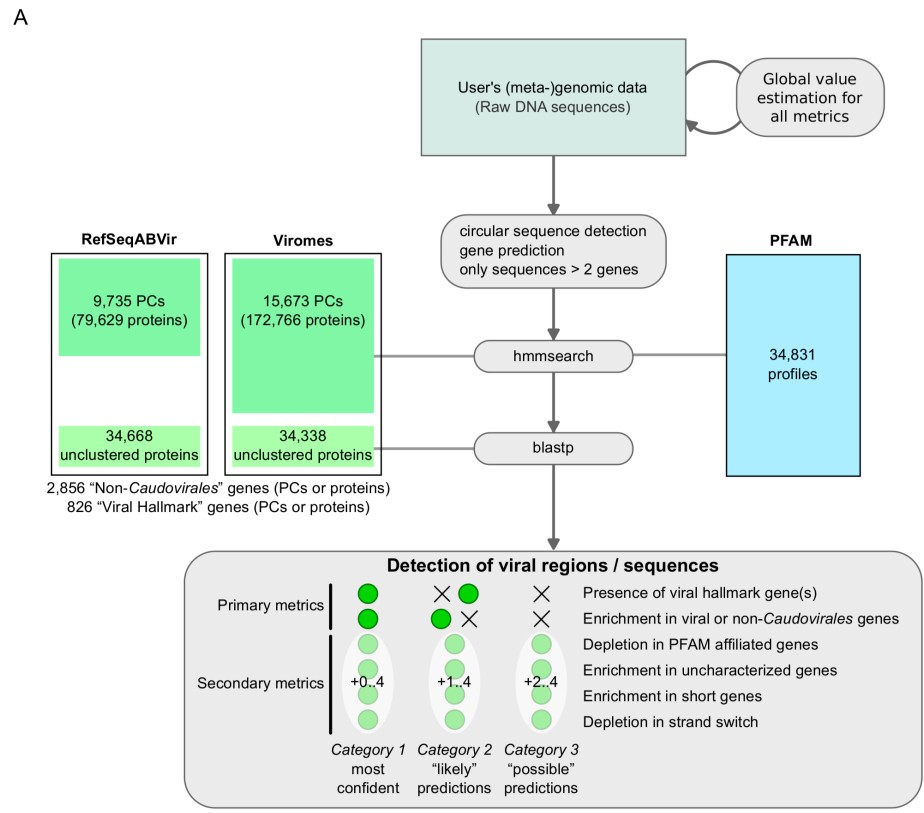

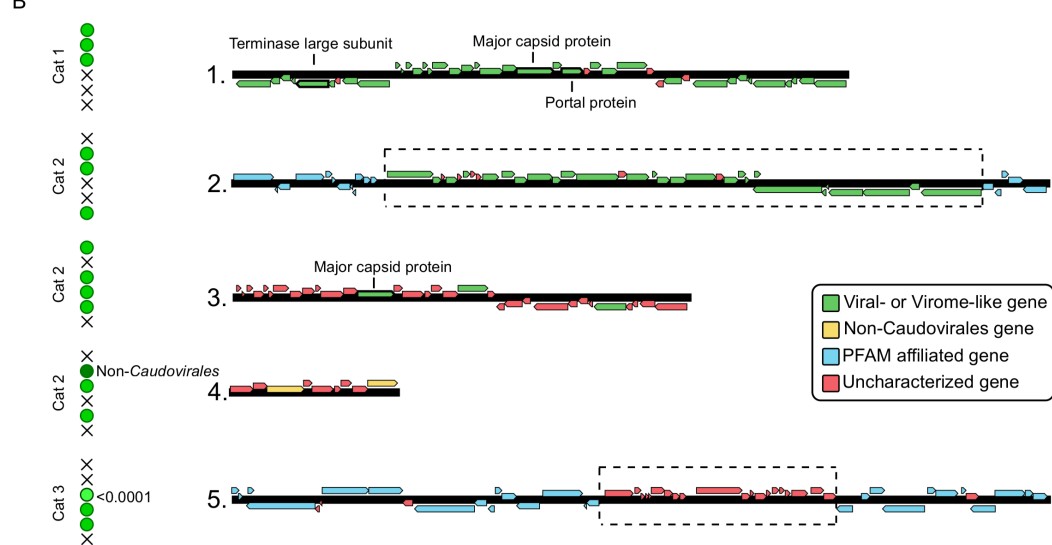

**Figure 1** **VirSorter process: overview (A) and examples of viral sequence detection (B).** (A) Overview of VirSorter process. The top part described the different parts of the sequence analysis pipeline, and the bottom frame summarizes the classification in three categories of decreasing confidence based on the different metrics being significant (green dot) or not (black cross). Viral "hallmark" genes or protein clusters (PCs) were identified by looking for genes typically of viral 

**Figure 1 (...continued)**
origin that are annotated as "major capsid protein," "portal," "terminase large subunit," "spike," "tail," "virion formation" or "coat" and manually removing all protein domains with a potential overlap with microbial functions. (B) Examples of viral sequence detection by VirSorter. On top is the clearest case, in which a sequence harbors several viral hallmark genes as well as enrichment in viral-like genes (or virome-like when the genes are most similar to a viral metagenome sequence, when using the Viromes database). This type of detection is considered as the most confident. The three examples below are different cases in which only one of the primary metrics is significant. Notably, these examples display how VirSorter can detect new viruses based on a significant depletion in characterized genes associated with a viral hallmark gene (case 3), and how the same number of genes can be a non-significant enrichment when considering all viruses, yet significant when looking at only the non-*Caudovirales* (case 4). These detections are still considered confident, although less sure than case 1. Finally, a last example (case 5) displays a more ambiguous situation, in which a sequence displays only secondary viral metrics but neither viral gene enrichment nor a viral hallmark gene. For these detections, one of the metrics (at least) must have an E-value lower than $10^{-04}$ (note that significance scores used in VirSorter output files are computed as negative $\log_{10}$ transformations of E-values, and would here correspond to a score of 4 or more).

databases, the VirSorter App on iPlant allows users to input their own reference viral genome sequence already assembled or to-be assembled using iPlant Apps prior to analysis with VirSorter. Assembled sequences are processed as follows: (i) genes are predicted with MetaGeneAnnotator (*Noguchi, Taniguchi & Itoh, 2008*), (ii) predicted proteins are clustered with sequences from the user-selected database (either RefSeqABVir or Viromes), and (iii) unclustered proteins are added to the "unclustered" pool. VirSorter scripts are also available through the github repository https://github.com/simroux/VirSorter.git.

## Comparison of VirSorter with other prophage predictors

We first evaluated VirSorter results against the manually curated prophages from (*Casjens, 2003*). Each genome was processed with VirSorter, PhiSpy (*Akhter, Aziz & Edwards, 2012*), Phage_Finder (*Fouts, 2006*) and PHAST (*Zhou et al., 2011*). For each tool, a prophage was considered as "detected" when a prediction covered more than 75% of the known prophage. For a more detailed example case of prophage detection in a complete bacterial genome including both prophages and genomic islands, the same tools were applied to the manually annotated *Pseudomonas aeruginosa* LES B58 genome (*Winstanley et al., 2009*).

VirSorter was then compared with the same prophage detection tools on the set of simulated SAGs. In that case, a viral sequence was considered as detected if predicted as completely viral or as a prophage. All the additional detections were manually checked to verify if the region was indeed viral (originating from a prophage in one of the microbial genomes rather than from a viral genome) or a false positive. The same approach was used for the simulated microbial and viral metagenomes results.

For each set of predictions, two metrics are computed. First, the Recall value corresponds to the number of viral sequences correctly predicted divided by the total number of known viral sequences in the dataset, and reflects the ability of the tool to find every known viral sequence in the dataset. Second, the Precision value is computed as the total number of viral sequences correctly predicted divided by the total number of viral sequences predicted, and indicates how accurate the tool is in its identification of viral signal.

## Simulation of draft genomes and metagenomes

A total of 10 Single-cell amplified genomes, 10 microbial metagenomes and 10 viral metagenomes were simulated with NeSSM (*Jia et al., 2013*). Microbial genomes were randomly picked within the bacterial and archaeal genomes available in RefSeq and WGS (as of January 2014). Viral genomes were picked within the most recently submitted genomes (since June 2014), thus are not in VirSorter reference database. Simulated inputs for each genome group (viral and microbial) followed a power-law distribution of abundances within the microbial and viral communities. The proportion of viral reads varied from 5 to 20% for microbial metagenome, and from 75 to 99% for viral metagenomes (Tables S4 and S5). For each simulated dataset, 100bp paired-end reads similar to those obtained with HiSeq Illumina were generated (100,000 for SAGs, 1,000,000 for metagenomes), QC'd with fastq_quality_trimmer with a threshold of 30 (part of the fastx_toolkit, http://hannonlab. cshl.edu/fastx_toolkit/), and assembled with Idba_ud (*Peng et al., 2012*).

To identify viral sequences in the assemblies, the resulting contigs were compared to the viral genomes with nucmer (*Delcher, Salzberg & Phillippy, 2003*), and all sequences matching one of the viral genomes at 97% nucleotide identity or more were considered as viral. All simulated contigs and composition table (i.e., relative abundance of each genome in the simulated dataset) are available in the iPlant Discovery Environment alongisde VirSorter results for each of these simulated datasets (/iplant/home/shared/imicrobe/VirSorter/Benchmark_datasets and Benchmark_results respectively).

# RESULTS & DISCUSSION

## Reference-dependent and general genome features used to detect viruses

VirSorter is designed to predict viral sequences in complete or fragmented genome sequence data from bacteria and archaea. Viral sequences are identified through a combination of "primary metrics" linked to the detection of significant similarities with known viral sequences and "secondary metrics" associated with viral-like genome structure (Fig. 1A). VirSorter first builds a probabilistic model for each metric using the microbial genomic data provided by the user (i.e., the complete genome or the entire contig dataset for draft genomes or metagenomes) that is then used as reference to calculate enrichment/depletion statistics. A "statistical enrichment in viral gene content" for a set of genes thus indicates that the region evaluated displays more viral-like genes than would be expected by chance alone based on the overall frequency of viral-like genes in the whole dataset. Viral-like genes are identified through comparison to RefSeq viral genomes ("RefSeqABVir" database hereafter) or to a custom database built from RefSeqABVir to which curated virome datasets were added to improve novel virus detection capabilities (hereafter "Viromes" database, Fig. 1A and Table S1). Through the VirSorter application (App) on iPlant, users can also add their own viral genome sequence(s) (in fasta format), which predicted protein will be added to the user-selected database (either RefSeqABVir or Viromes).

## Viral signal mining process

Viral regions are predicted based on a summary of primary and secondary metrics evaluated on each genomic sequence. Each prediction is categorized from 1 to 3 in order of decreased confidence (Figs. 1A and 1B). Sequences for which the predicted viral region spans more than 80% of the contig length are considered as entirely viral. Biologically, we interpret these different categories as sequences similar to known viral references (*category 1*), sequences divergent from references with mostly genes yet to be detected in viral genomes or partial sequences lacking viral hallmark genes which may include defective prophages (*category 2*), and sequences or regions with a genome structure similar to viral genomes, but lacking any similarity to known viruses or viromes (*category 3*). These latter, *category 3* predictions are thus essentially "aberrant" cellular genomic regions, and as such should be carefully examined as this category also routinely includes hypervariable microbial genomic islands and other mobile genetic elements in addition to novel viral sequences. However, we include *category 3* predictions since when coupled to manual inspection, researchers can use these predictions to uncover novel biology, particularly when analyzing the small contigs and highly novel viruses likely to derive from fragmented draft genomes or SAGs.

## Virsorter prophage prediction is comparable to existing tools

To evaluate VirSorter performances, we first examined its prophage prediction capability as compared to existing tools. Specifically, we used a set of 267 manually annotated prophages from 54 bacterial genomes (*Casjens, 2003*) to compare the prophage prediction performances of VirSorter, PhiSpy, Phage_Finder, and PHAST. We evaluate performance using two metrics: (i) "Recall," the number of viral regions detected divided by the total number of viral regions (also known as "Sensitivity") and (ii) "Precision," the number of correct predictions divided by the total number of predictions (also known as "Positive Predictive Value").

All of the tested prophage prediction tools perform well on these complete genome datasets as Recall values range from 64 to 85%, and Precision values range from 74 to 93% (Fig. 2 and Table S2). Two of the tools also associate their predictions with a level of confidence: PHAST predictions are noted as "intact," "incomplete," or "questionable" based on the number and type of phage genes detected, and VirSorter categorizes predictions as described above. To see how these confidence categories impacted results, we computed scores with and without the least confident predictions for both of these tools (Fig. 2). For PHAST, adding the questionable detections increased detection sensitivity (Recall increased from 70 to 84%) without altering the Precision (both sets of predictions display a Precision of 83%). Conversely, including the least confident *category 3* predictions for VirSorter only slightly increased Recall (73 to 79%), but did so at the cost of Precision (dropping from 93 to 72%). Hence, for VirSorter, prophages predicted as *category 3* from complete microbial genomes are prone to "false-positive" detections—notably because they can also include other genomic regions with unusual sequence composition features such as genomic islands or mobile genetic elements (see below).

**Peer**J

A

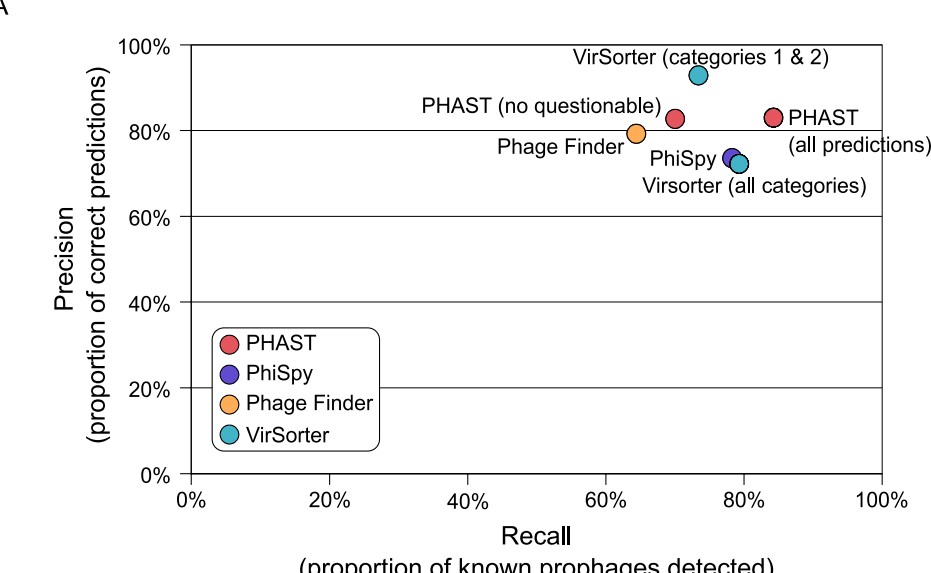

B

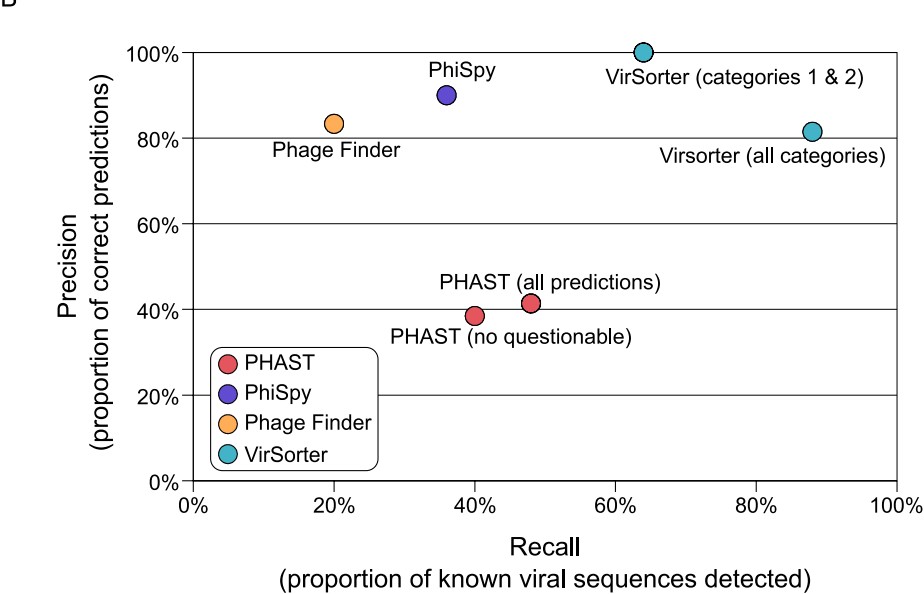

**Figure 2** **Accuracy of viral sequence predictions of VirSorter, PHAST, Phage-finder and PhiSpy on (A) complete microbial genomes, and (B) draft genomes from simulated SAGs including a microbial and viral genome.** For each set of predictions (i.e., each tool and set of option when applicable), the two metrics used to evaluate the tool performance are Recall (*x*-axis, proportion of known viral sequences or regions detected) and Precision (*y*-axis, proportion of predictions that corresponded to known viral sequences or regions). Prophages identified in the complete microbial genomes are compared to the list of manually curated prophages from *Casjens (2003)*.

**Table 1 Comparison of VirSorter predictions with prophage predictors on *Pseudomonas aeruginosa* LES B58 genome (NC_011770).** The coordinates of each prophage known on *Pseudomonas aeruginosa* LES B58 genome and detection for the different tools are indicated, with absence of detection highlighted in red. For VirSorter and PHAST, the category of detection (1, 2 or 3 for VirSorter, intact, incomplete or questionable for PHAST) is also indicated. False-positive detections of genomic islands as putative prophages are highlighted in orange.

| Feature | Coordinates | VirSorter | PHAST | PhiSpy | Phage_Finder |
|---|---|---|---|---|---|
| Prophage 1 | 665,272–680,608 | Prophage – 2 | Prophage – questionable | Prophage | – |
| Prophage 2 | 863,875–906,018 | Prophage – 2 | Prophage – questionable | Prophage | Prophage |
| Prophage 3 | 1,433,756–1,476,547 | Prophage – 2 | Prophage – questionable | Prophage | Prophage |
| Prophage 4 | 1,684,045–1,720,850 | Prophage – 2 | Prophage – questionable | Prophage | Prophage |
| Genomic Island 1 | 2,504,700–2,551,100 | Prophage – 3 | Prophage – questionable | – | – |
| Prophage 5 | 2,690,450–2,740,350 | Prophage – 1 | Prophage – intact | Prophage | Prophage |
| Genomic Island 2 | 2,751,800–2,783,500 | – | – | – | – |
| Genomic Island 3 | 2,796,836–2,907,406 | – | – | Prophage | – |
| Genomic Island 4 | 3,392,800–3,432,228 | – | – | – | – |
| Prophage 6 | 4,545,190–4,552,788 | Prophage – 2 | Prophage – intact | – | – |
| Genomic Island 5 | 4,931,528–4,960,941 | Prophage – 3 | – | – | – |

Next, we focused on the case of prophages prediction in the manually annotated *Pseudomonas aeruginosa* LES B58 genome, which includes both prophages and genomic islands (*Winstanley et al., 2009*), to better explore how these tools deal with divergent prophages and unusual genomic regions (Table 1). All 6 known prophages in this genome were detected by VirSorter (categories 1 or 2), and PHAST (though 4 were considered "questionable"), whereas PhiSpy and Phage_Finder detected only 5 and 4, respectively. These missed prophages were the shortest ones (12 and 19 genes compared to >40 genes for all the other prophages), and one (Prophage_6) also corresponded to an unusual phage from the *Inoviridae* familiy, under-represented in viral genome databases. Beyond prophages, this microbial genome also displayed 5 manually curated genomic islands. None of these genomic islands were detected as a prophage by Phage_Finder, while PhiSpy and PHAST each wrongly identifies one of these genomic islands as a prophage, and VirSorter identifies two of them as *category 3* predictions (i.e., putative prophage or other unusual genomic feature, Table 1). This example illustrates that *category 3* predictions from VirSorter help capture even divergent prophages, but also detect hypervariable regions in microbial genomes, such as genomic islands or plasmids.

## VirSorter is more efficient at mining viral signal from single-cell amplified genomes (SAGs)

To evaluate the capacity of prophage predictors and VirSorter to detect viral sequences in SAG datasets, we generated 10 simulated datasets of 100,000 reads (100bp) from one microbial and one viral genome, with 5 to 10% of the reads originating from the viral genome (Table S3). For each simulated dataset, reads were assembled into contigs (averages = 556 contigs per SAG ~3.3 kb in length), from which viral sequences or prophages were then predicted (the viral genomes used in the simulated datasets being absent from the VirSorter reference database).

On these SAGs, VirSorter outperformed all other tools as the only one maintaining comparable Recall and Precision values to those from complete microbial genomes (Fig. 2B). VirSorter *categories 1* & *2* (higher confidence predictions) displayed a Recall of 65% and a Precision of 100%, while adding in category 3 predictions increased Recall (88%) but reduced Precision (81%). Thus, for fragmented genomes, *category 3* predictions help recover more viral sequences, but do so at the cost of increased false-positives. In comparison, PHAST (with or without the "questionable" predictions) performed at 40–50% Recall and 38–41% Precision, whereas PhiSpy and Phage_Finder had a lower Recall (36 and 20%, respectively) but high Precision (90 and 83%, respectively). Considering that the prophage detection tools were optimized for viral sequence detection in complete microbial genomes, it is not surprising that VirSorter performs better for fragmented genomes.

We also applied VirSorter and the prophage predictors to a set of 127 SAGs from the uncultivated bacteria SUP05 for which viral sequences were previously manually identified and curated (*Roux et al., 2014b*). Of the 69 known viral contigs in this dataset, 62 were detected by VirSorter (including 29 as *category 3*), with the remaining 7 being too short (5.1kb on average) to provide significant enrichment scores. In contrast, PHAST, PhiSpy and Phage_Finder detected only 15, 1 and none of these sequences, respectively. Beyond the fragmented nature of these SUP05 SAGs, these data likely represent a worst case scenario for the prophage prediction tools as these 69 SUP05 viruses represented new viral genera, and thus no closely related reference sequence were available in databases (*Roux et al., 2014b*).

## VirSorter alone is able to mine viral signal from bacterial and viral metagenomes

Next, we evaluated VirSorter's capability to recover viral sequences in fragmented genomes assembled from metagenomic datasets. To this end, we created 10 'metagenomes' from 15 microbial and 15 viral genomes at varying representative abundances (Table S4 and see 'Methods'). These simulated datasets total 192,941 contigs, so the scale is quite large— none of the prophage predictors were able to even process the data in a reasonable time (i.e., less than several days). Given that metagenome-derived contigs also represented fragmented genomes, we expect that performance would have been poor for prophage prediction tools on these datasets—likely even worse than the SAGs performance testing above.

VirSorter, however, was designed for and is thus able to handle such datasets. For contigs greater than 500bp, VirSorter predictions displayed good Precision (93–100%) but low Recall (33%, Fig. 3A and Table S4). However, as the size of the contigs increased, Recall increased to 79–84% for contigs >3kb and 95–97% for contigs >10kb, with no Precision loss (Fig. 3B).

Finally, we evaluated the potential of VirSorter to detect viral sequences in viral metagenomes contaminated with cellular genomes. Such cellular sequence in viral-fraction metagenomes can derive from co-purified encapsidated DNA (in gene transfer agents or generalized transducing phages) or contamination, and represents a common challenge in making inferences from viromes (*Roux et al., 2013*). We thus simulated 10 viral

A

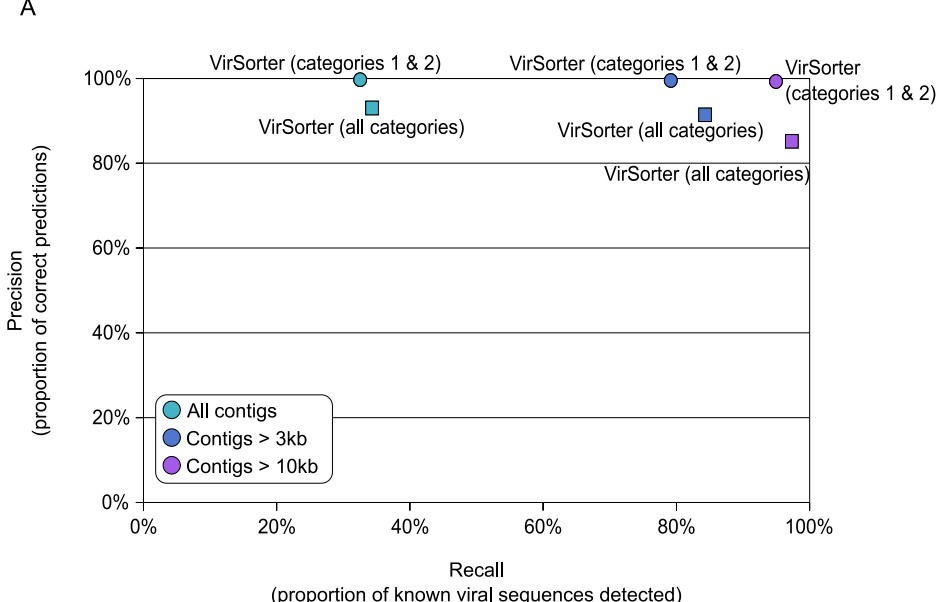

B

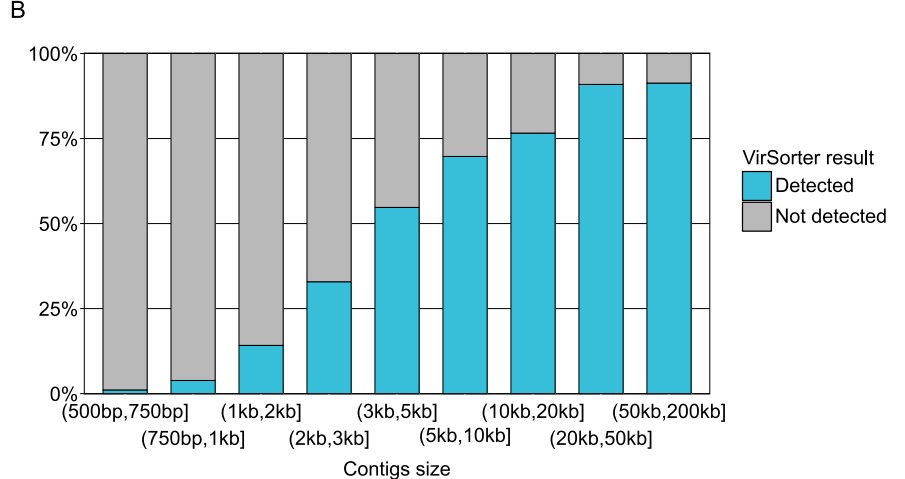

**Figure 3 Detection of viral sequences in microbial metagenomes by VirSorter.** (A) Average Recall (*x*-axis) and Precision (*y*-axis) of viral sequence detection by VirSorter in 10 simulated microbial metagenomes for different contig size thresholds. (B) Detection of viral sequences by VirSorter in simulated microbial metagenomes by contig size fraction.

metagenomes of 1,000,000 reads (100bp) from a mix of 15 microbial and 60 viral genomes. This time, we simulated metagenomes where viral reads represented a larger proportion of the dataset, ranging from 75 to 99% (Table S5). Here, all microbial genomes available in RefSeq and WGS (as of January 2014) were used by VirSorter to model microbial genomic metrics instead of the whole dataset, since viromes largely lack microbial sequences. This usage case of VirSorter is implemented in the iPlant application and is available by checking the box "virome decontamination" in the submission form.

**Table 2 Results of VirSorter viral sequence detection on simulated viral metagenomes with a limited contamination by cellular genomes (1 to 25% of raw reads).** Metrics presented are Recall (proportion of viral sequences detected) and Precision (proportion of predictions corresponding to viral sequences).

| | VirSorter—categories 1 & 2 | | VirSorter—all categories | |
|---|---|---|---|---|
| | *Recall* | *Precision* | *Recall* | *Precision* |
| **All contigs (>500bp)** | 31.71% | 99.89% | 32.96% | 99.79% |
| **Contigs >3kb** | 85.64% | 99.80% | 90.29% | 99.62% |
| **Contigs >10kb** | 97.14% | 99.48% | 99.82% | 98.99% |

As found above for prediction of viral sequence data from the microbial metagenome simulations, VirSorter performance as a 'virome decontaminator' improves as contig size increases (Table 2 and Table S5). When considering all contigs (>500bp), the Recall of viral sequences is 32% on average, but increases up to 86% for contigs >3kb and 97% for contigs >10kb. When *category 3* predictions are included, these Recall values increase slightly to 33%, 90% and 99.8% for the increasing contig sizes, respectively. At the same time, the Precision of viral sequence detection stays high for all contig sizes, even when including *category 3* predictions (99% and more, Table 2).

## VirSorter's strengths and weaknesses

VirSorter represents a novel, scalable, and community-available tool for detecting and identifying viral genome sequences from diverse microbial datasets. Its performance for prophage prediction is largely comparable to that of available prophage prediction tools when applied to complete microbial genomes, but it outperforms available tools when making predictions from modern microbial datasets which tend to be fragmented and larger-scale or when searching for viruses beyond those "known" in current databases. Thus, VirSorter complements existing tools to help elucidate bacterial and archaeal viral sequences among myriad modern microbial genomic data types.

However, VirSorter does have limitations. First, VirSorter was designed and optimized for detection of bacterial and archaeal viruses, so it does not detect eukaryotic viruses well. This is because the database lacks eukaryote viruses, and the viral genome features were only evaluated on bacterial and archaeal viruses. VirSorter will still detect eukaryote viruses, often as *category 3* because of their singular genome composition (compared to a typical cellular genome), but its capacity is extremely limited in its current build. Second, short (<3kb) viral contigs will tend to only be detected by VirSorter when they contain hallmark genes. Pragmatically, this means that viral signal detection in non-assembled reads or in contigs assembled from (meta)transcriptome data will usually be inefficient. Third, prophage prediction tools also look for additional signs of prophages such as the presence of integrase genes, *att* sites, or repeat features to demarcate the 'ends' of a prophage genome, none of these features are examined by VirSorter. Thus, prophage prediction tools likely remain the best means to most accurately annotate prophages in a complete microbial genome, whereas VirSorter is best used for high-throughput analyses and for detecting viral signal in fragmented genomes. Finally, *category 3* detections repre-

sent sequences and regions that are unique within the genome(s) being compared, so while many can be viral, these predictions could also represent other mobile genetic elements or hypervariable genomic islands and require manual curation. The only case where *category 3* predictions may be considered without manual curation are viral metagenome decontaminations as these predictions increase Recall while only marginally lowering Precision.

## ACKNOWLEDGEMENTS

We thank Nirav Merchant, Darren Boss, and Ken Youens-Clark for aiding in setting up VirSorter on the iPlant platform, Rachel Whitaker and Whitney England for their assistance with *Pseudomonas aeruginosa* LES B58 genome annotation, as well as TMPL members for comments on the manuscript.

### Funding

This work was performed under the auspices of the Gordon and Betty Moore Foundation (#3790) through grants awarded to Matthew B. Sullivan. Simon Roux was partially supported by the University of Arizona Ecosystem Genomics Institute through a grant from the Technology and Research Initiative Fund through the Water, Environmental and Energy Solutions Initiative. The funders had no role in study design, data collection and analysis, decision to publish, or preparation of the manuscript.

### Grant Disclosures

The following grant information was disclosed by the authors:
Gordon and Betty Moore Foundation: #3790.
University of Arizona Ecosystem Genomics Institute.

### Competing Interests

The authors declare there are no competing interests.

### Author Contributions

- Simon Roux conceived and designed the experiments, performed the experiments, analyzed the data, wrote the paper, prepared figures and/or tables, reviewed drafts of the paper.
- Francois Enault conceived and designed the experiments, wrote the paper, prepared figures and/or tables, reviewed drafts of the paper.
- Bonnie L. Hurwitz performed the experiments, wrote the paper, prepared figures and/or tables, reviewed drafts of the paper.
- Matthew B. Sullivan conceived and designed the experiments, analyzed the data, wrote the paper, prepared figures and/or tables, reviewed drafts of the paper.

### Supplemental Information

Supplemental information for this article can be found online at http://dx.doi.org/10.7717/peerj.985#supplemental-information.

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
