# Peer review of "VirSorter: mining viral signal from microbial genomic data"

_PeerJ, doi:10.7717/peerj.985_

## Round 0.1 · original submission · Minor Revisions

· Academic Editor

Minor Revisions

I agree with both reviewers as to the novelty and utility of VirSorter. Particularly, the abilities of this tool to detect novel viruses from metagenomic or fragmented sequences, to detect extrachromosomal prophages, and to perform 'virome decontamination' are very useful features. Combining these features with a web interface is fantastic, as otherwise, there are many researchers who would find the tool too difficult to use.
Additionally, this manuscript is overall well written and pleasurable to read. However, there are some minor issues to be dealt with before this manuscript is suitable for acceptance. Both reviewers have listed some minor comments, which I would like you to address, including the request for some supplementary information that clarifies how to use the software. You could either provide this "how to" information as a supplementary file or in the manuscript you could point users to an online tutorial. Furthermore, below are some minor editorial comments of my own that I would like you to address:
1. It appears to me that line 103 may be missing a word or otherwise could be reworded for clarity.
2. Line 223- perhaps 'microbial genome' should be plural.
3. Line 296- "..from 74 and 93%.." should read "..from 74 to 93%.."
4. Line 347 "..represents.." should be "..represent.." since data is plural.
5. Line 410- "predictions" is misspelled.
6. Line 419- species names should not be capitalized.
7. Figure 1- in the blue pfam box, it appears that a comma is missing.
8. Please examine figure legends for consistency. There are some inconsistencies in whether part A and B titles are bolded.
9. In the title of Table 1, genus and species name should be italicized.

Reviewer 1 ·

Basic reporting

This manuscript describes VirSorter, a much-needed tool in the field of viral genomics and ecology. The amount of microbial genome sequence data available in public databases is increasing at a rapid pace, yet few tools exist that enable users to mine the viral signal from these large datasets. Based on all the validation work presented in this manuscript (see my positive comments in Experimental Design section), the VirSorter tool appears to represent a significant advance compared to available tools and in particular, it fills a distinct niche in that it can work on highly fragmented genomic sequence data (such as SAGs or metagenomic contigs).

Minor, specific comments:

The term “viruses of microbes” needs to be defined, as this can include viruses that infect bacteria, archaea, and small eukaryotes. As the authors mention in the discussing, this tool would not be expected to perform well for eukaryotic viruses, since they were not included in the databases used to develop the tool. I’m assuming the authors chose to talk about “viruses of microbes” instead of “phages” because archaeal viruses have also been included in the databases. I think it is fine if the authors choose to use this term, but it should be defined at first use (phages and archaeal viruses).

Last sentence of abstract: the term "non-targeted databases" is unclear here. I assume it means microbial metagenomes or SAGs or things other than "viral-specific databases" - but maybe change the wording to reflect that.

Line 33: Supplement these references with additional references from other lab groups. For example, Thompson et al. PNAS 2011. Also, I’m surprised that the authors did not use the term “auxiliary metabolic genes” to refer to these genes. It seems like this tool could be used to identify host genes within viral contigs as well.

Line 54: The statistic that temperate viruses are present in 60% of sequenced bacteria and archaea is misleading. To the best of my knowledge (and certainly according to the 2 references cited here), temperate viruses are not prevalent in archaeal genomes. Please clarify.

Lines 59-61: A lot of the pioneering work in this field of identifying viral sequences amongst microbial metagenomes and then linking metabolic genes found on those viral contigs was done by Oded Beja’s group. Some references should be included in the introduction.

Line 66: Add the reference for PHAST here (it is included later in the paper but missed here)

Line 268: Typo in this sentence

Line 299: “categorize” should be “categorizes”

Line 328: “datasets” should be “dataset”

Lines 363-367: These sentences are redundant, I think you meant to delete one of them.

Line 378: Typo “theAs”

Line 443: should be “in revision”

Line 518: Pride reference has problem with author middle initial caps.

References have inconsistencies in journal title capitalization.

Table S1 legend: Bold section needs rewording, confusing as written.

Experimental design

The experimental design of building the VirSorter tool, as well as the comparative analyses with other publicly available programs on actual sequence data as well as simulated data is extremely thorough and well-done. I especially appreciate that conserved domains (e.g., protease, chaperone) were removed to avoid false positive results, and the iterative process to refine the sequence space reduces the problems caused by limited reference genomes. I found the tool comparisons easy to understand and they clearly highlight the strengths and limitations of this new tool, as well as providing guidance to readers on when this tool would be most useful.

Minor, specific comments
Line 90: Are there plans to update the “RefSeqABVir” database on some sort of regular basis (e.g., annually) to keep this tool at the cutting-edge as more genomes are sequenced?

Line 145-146: In “(iii) depletion in PFAM affiliated genes”, does that exclude viral PFAMs?

Validity of the findings

All conclusions are valid. This manuscript could be enhanced by providing screenshots of the tool output as supplementary information, because from reading, I have no idea what the actual raw output from this tool looks like.

Additional comments

I am not familiar with the Discovery Environment of the iPlant Cyberinfrastructure, and when I googled the VirSorter tool to try it out for myself while conducting this review, I could not figure it out. I found a wiki for the tool (https://pods.iplantcollaborative.org/wiki/display/DEapps/VIRSorter+1.0.1) but couldn't figure out how to start using it. Using the link provided in the manuscript ((https://de.iplantcollaborative.org/de/) I was able to create an iPlant account and start working with the program, but I feel like this will be a sticking point for many other viral ecologists and may limit use of this program (compared to something like MetaVir which has a direct website). Please provide more information to the reader about the iPlant environment, including a tutorial (maybe as supplemental) for how to do the basics, like important files for analysis.

·

Basic reporting

The article is clearly written and readable without sinking into a morass of figures. The tenor of writing is appropriate and engaging without being colloquial. The scope of the work is well-demarcated, being a description and comparison of one tool against a handful of other tools.

Experimental design

The metrics used to measure VirSorter against PhiSpy, Phage_Finder, and PHAST are appropriate for comparison, and effort was made to normalize the results to allow for comparison. The normalization methods are well-described, as well as the basic algorithmic choices made in the program. In this case, a brief description is appropriate, since the tool and its code are available to the reader.

Lines 142-149: The multi-pronged approach shown is prudent. Dinucleotide bias could be included as a metric, but I don't believe it would meaningfully improve the performance of the tool.

The method used by the program is clearly defined, and the comparison metrics are meaningful across tools.

Validity of the findings

The data as presented are included in the supplemental tables, and appear to be consistent with my knowledge of the tools. The conclusions are appropriate and conservative given the significant superiority of this tool for the specific use case of high-throughput prophage screening.

Additional comments

1. As a biodefense researcher, this method would be of great interest to the defense community if expanded to cover RNA viruses. You plan to address this in the future, and I'm excited.
2. Line 235: Capitalize January
3. Line 337: Please keep Recall and Precision in the same order, as a favor to the reader.
4. Line 368: The reversed use case is an excellent tool in and of itself. Very few methods exist to clarify viral noise, this could well be a standard.
5. Lines 320 and 408: Category 3 is at serious risk of generating fishing expeditions, especially when the user has already decided what they want to find. You've done well to include a clear warning against trusting the dubious hits blindly.

---

## Round 0.2 · accepted · Accept

· Academic Editor

Accept

Thank you for addressing the specific comments made by the two reviewers and myself, especially the addition of the supplementary figure showing how to use the tool.